# Construction of management tools for early warning of prevacuum steam sterilization failure

Xin Zhao[1], Ting Liu[2], Ning Ma[1], Ran Wang[2], Ying Li[2], Tong Shen[2]*

**1** Central Sterile Supply Department, Xuan Wu Hospital Capital Medical University, Beijing, China, **2** Central Operating Room, Xuan Wu Hospital Capital Medical University, Beijing, China

* xwyy18618247182@163.com

## Abstract

### Objective

The aim of this study is to develop a set of management tools for early warning of steam sterilization failure, including a failure risk checklist and a calculation model for assessing early sterilization failure risk, and to verify the early warning effectiveness of the management tools through check experiment.

### Methods

This study included two stages. The first stage involved the construction of a failure risk checklist and the development of a computational formula for early sterilization failure risk by expert consultation. In the second stage, the early warning effectiveness of the sterilization failure management tools was verified through a comparative verification method based on the consultation results of two experts.

### Results

The evaluation system comprised a total of 10 early warning indicators divided into 3 categories: "Staff", "Steam sterilizer" and "Medical devices". The Cronbach's α coefficient for internal consistency reliability was 0.837, the total content validity index (S-CVI) was 0.884, and the item-content validity (I-CVI) ranged from 0.762 to 1. In management practices, the sterilization qualification rate of the experimental group was 100%, compared to 99.78% in the control group. There were significant differences between the two groups in terms of accidental downtime (hours) and the normal operation rate of the equipment caused by equipment failure.

### Conclusion

This study constructed an early warning model for the vacuum steam sterilization process failures. This model facilitates more accurate identification of sterilization failure risks, which helps medical institutions in enhancing sterilization quality control and early interventions.

**Data Availability Statement:** "***AT ACCEPT: Please follow up with the authors to share repository details for uploaded data*** All relevant data are within the paper and its Supporting Information files."

**Funding:** The author(s) received no specific funding for this work.

**Competing interests:** The authors have declared that no competing interests exist.

## Introduction

The quality of sterilization directly affects the safety of sterile packages, and is crucial for the life and health of patients. Inadequate sterilization of medical devices can have severe consequences for patients and can also result in significant financial and reputational costs for manufacturers and healthcare facilities [1]. The prevacuum steam sterilizer was first developed in the UK in the 1950s. Due to its shorter sterilization time, enhanced sterilization effectiveness, higher working efficiency and greater energy consumption than lower exhaust sterilizers, the pre-vacuum steam sterilizer has become a vital component in the reprocessing process of reusable medical devices in the sterile supply center [2, 3]. The pressure steam sterilizer can achieve sterilization by repeatedly extracting vacuum and filling steam into the sterilization room, causing the sterilization room to reach a certain vacuum, and then filling saturated steam to achieve the desired pressure and temperature while maintaining the set time. The sterilization temperature, pressure, time, and other process parameters have a significant impact on the sterilization effect. Each sterilization procedure should be continuously monitored, with essential parameters like temperature, pressure, and time recorded [4].

Researches have identified several key factors that were associated with sterilization failure: the absence of strict regulatory requirements, lack of appropriate instructions, lack of supervision, power failures, inadequate knowledge, improper sterilization temperature and holding time, incorrect packaging and loading procedures, malfunctioning equipment, and inadequate maintenance of equipment [5–7].

In order to ensure a safe, effective and controllable sterilization cycle, manufacturers of sterilizers have incorporated various fault alarm checkpoints and object sensors tailored to the specific characteristics of their devices to monitor the chamber parameters of the sterilization process and prevent sterilization failure. With the help of these checkpoints and detectors, sterilization operators also increasingly rely on the alarm mechanism of the sterilizers to assess whether the sterilization process is safe and effective [8]. Once the checkpoint and sensor detect abnormalities, the entire sterilization process is failed, leading to negative consequences such as the need for reprocessing sterilized devices and surgery delays. Despite implementing numerous preventive measures, issues such as wet packs, excessive colony counts and infections associated with sterilized devices continue to occur [9].

The CSSD is typically in charge of daily inspection, cleaning, and disinfection as part of the pressure steam sterilization equipment management and sterilization process monitoring process. When the pressure steam sterilization equipment fails, CSSD should contact the manufacturer to arrange for on-site servicing. It takes a long time to diagnose a malfunction, incurs considerable maintenance costs, and causes significant equipment delivery delays. Therefore, it is extremely vital to avoid beforehand [10]. Currently, research on sterilization process failure is primarily based on retrospective analysis of failure data from pressure steam sterilizers in hospitals, and there is a lack of preventive methods and tools that can control the risk of failure in advance and reduce the negative impact of maintenance costs and medical services caused by sterilization failure.

In this study, we established an early warning model including a failure risk checklist and a dynamic surveillance method to improve the prediction ability for sterilization failures and to reduce the health-related economic losses.

## Methods

### Research design

This study aimed to develop a more comprehensive warning method for failures in prevacuum pressure steam sterilization. The method was divided into two parts. First, the failure risk

checklist was created through multidisciplinary expert consultations, which is intended for standardized inspections and early interventions of the entire sterilization process and related influencing factors. The second part involved a calculation model that incorporates process control methods from the industrial manufacturing field. This model continuously monitors and analyzes the readings of temperature and pressure sensors to detect early sterilization failure. The improved method facilitates the early detection and intervention of sterilization failures, ultimately aiming to reduce the economic losses incurred from sterilization failures.

The data used in this study came from the data generated by the sterilization process of CSSD, and did not involve the personal information of patients and the diagnosis and treatment process. This study was exempted by the ethics committee and participant consent was waived by the ethics committee of Xuanwu hospital.

## The Delphi method

We firstly established an expert group comprising 5 staff from the CSSD, 6 steam sterilizer engineers, 5 experts of nursing management and 2 specialists in hospital-acquired infection management. The selection criteria for experts are active engagement in the sterile supply or steam sterilizer maintenance for over 5 years, and possession of a bachelor's degree or higher.

The purpose of the expert group was to achieve a multidisciplinary consensus on the failure risk checklist and the calculation model for early sterilization failure risk, through multiple rounds of questionnaire consultations.

For the first part, the failure risk checklist was created based on the findings of previous studies. Two experts of the research group screened the PubMed, MEDLINE, and WANFANG Med online databases with terms "steam sterilization", "sterilization quality", "CSSD", "sterilization failure" and "disinfection" to review literature for influencing factors related to sterilization failure. The inclusion criteria for articles were as follows: 1) addressed quality control for prevacuum steam sterilization process or analyzed sterilization failures; 2) published between 1st January 2010 to 31st December 2023; 3) written in English or Chinese; and 4) full-text availability. Articles were excluded if: 1) they were comments, case reports, editorials, letters, or conference summaries; 2) unable to find full English texts.

For the second part, the calculation model for assessing early sterilization failure was designed by the expert group using questionnaires [11]. It referred to the *Technical Requirements for Large Steam Sterilizers* (GB 8599–2023) concerning sterilization variables for steam sterilizers larger than 800L as well as the safety margin theory in industrial design [12].

In this study, a two-round Delphi process was conducted to achieve the stated aims. In the first round, the experts were required to rate the quality improving measures on a five-point Likert scale based on their importance via a questionnaire. Indicators with scores above 3.5 were selected for the second round [13]. In the second round, the experts were asked to indicate "agree" or "disagree" with each indicator, with those achieving at least two-thirds agreement among experts included [13, 14].

The questionnaires for expert consists of three parts: the first part collected the basic information about the experts; the second included the main body (failure risk checklist and calculation model for early sterilization failure); and the third evaluated the expert authority. The experts assessed the early warning indicators according to their importance and accessibility [15]. Familiarity with early warning indicators was rated on a scale of 1 to 5 by "completely unfamiliar", "quite unfamiliar", "neutral", "quite familiar", and "strongly familiar" [16].

The expert authority (Ca) was determined through the evaluation index (Ci) and the familiarity of the evaluation index (Cs) using the formula Ca = (Ci + Cs) / 2. A Ca value of 0.70 or higher indicated high expert authority [17].

Cronbach's α was a common method to evaluate the reliability of questionnaires. A Cronbach's α coefficient ranging from 0.8 to 0.9 indicated meeting the internal consistency requirements. The content validity index for the item (Item-level CVI, I-CVI) was 0.78, scale-level content validity index (Scale-level CVI, S-CVI) was 0.8, suggesting that the evaluation system possesses good content validity [18].

## Management practice

The test object for this study was the steam sterilizer in the CSSD of the Xuan Wu Hospital of Capital Medical University. The steam sterilization processes were analyzed in two groups: the conventional group, which encompassed the sterilizations conducted between August 2021 and April 2022, and the validation group, which included the period from May 2022 to January 2023. During the validation group period, the failure risk checklist and calculation model were implemented during the steam sterilization process. The primary evaluation indices used to assess test effectiveness are alarm times, sterile quality of sterilized package and accidental downtime.

# Results

## The expert consultation

A total of 11 experts participated in the consultation, including 9 males and 2 females. The age distribution was as follows: 7 experts were aged 30–39 years,2 were aged 40–49 years and 2 were over 50 years; In terms of education, 8 had undergraduate degrees, 3 held master's degrees; Regarding their working experience in this field, 5 experts had 5–10 years of experience, 4 had 11–15 years, and 2 had 15 years; The distribution of professional titles included 7 with intermediate titles and 4 with deputy senior titles.

All 11 questionnaires were issued and recovered. The mean Ci was 0.851, mean Cs was 0.823, and the Ca was 0.837, all of which exceeded the general requirement of 0.7. The Cronbach's α coefficient for reliability evaluation was 0.863, the scale-level content validity index (S-CVI) was 0.884, and the item-level content validity (I-CVI) ranged from 0.762 to 1, which met the general requirements of questionnaire reliability and content validity. The failure risk checklist for prevacuum steam sterilization was in Table 1.

## Real time early failure surveillance

The input data was as follow:

1. $T_1$: Readings from Temperature sensor 1

2. $T_2$: Readings from Temperature sensor 2

3. $T_3$: The converted temperature value from pressure sensor 1

4. $T_4$: The converted temperature value from pressure sensor 2

    Formulas for calculating the safety margin of sterilization variables were as follows:

1. Sterilization temperature range safety margin = (3˚C-Measured sterilization temperature range)/3˚C×100%

2. Sterilization temperature uniformity safety margin = (Measured maintenance time-180s) /180s ×100%

3. Sterilization temperature uniformity safety margin = (2˚C-Difference of measured temperature uniformity)/2×100%

**Table 1. Failure risk checklist for prevacuum steam sterilization.**

| Categories | Indicators | measures |
|---|---|---|
| Staffs | Training of theoretical knowledge | 1. 20 hours of learning covering standards, regulations, operation processes, and department assessment<br>2. Passing the exams of theoretical knowledge<br>3. Summarizing 5 research papers on wet pack or other steam sterilization failure |
| | Training of sterilizer operation | 1. 40 hours of pre-job operation training, including standard operating procedures, emergency response procedures, et etc.<br>2. Operating practice for at least twice per year Practical operation experience is required at least twice a year |
| | Quality control team | 1. The team consists of the director of CSSD, packaging group leader, cleaning group leader, sterilization group leader and quality control group leader.<br>2. Holding a monthly quality analysis meeting to discuss workload, risk assessments and improvement measures |
| Steam sterilizer | Process monitoring | 1. Conducting regular physical, chemical and biological monitoring.<br>2. Using biological culture and chemical PCD to ensure sterilization effectiveness<br>3. Wet pack inspections and root cause analysis |
| | Equipment inspection | 1. Annual checks of steam quality<br>2. Steam pipeline inspections every 4 hours to detect any water leaks<br>3. Regular checks of alarm sensor sensitivity |
| | Equipment maintenance | 1. Regular maintenance<br>2. Sensor calibration is necessary when:<br>a. The pre-conditioning time is longer than or equal to 8 minutes and fail to reach the target pressure value<br>b. The two temperature sensors record temperatures lower than 134.2°C<br>c. The deviation between the two temperature sensors exceeds 0.6°C<br>d. The reading of the two pressure sensors exceed 60 mbar |
| Medical devices | Design of reprocessing procedure | 1. Setting appropriate parameters for pressure steam sterilization according to guidelines, standards, and specific requirements from manufacturers. |
| | Cleanliness and dryness | 1. Cleaning before packaging<br>2. Performing daily visual inspection, using a lighted magnification as needed<br>3. Ensuring instruments are dry before packaging. Using drying cabinets or an air gun to dry if necessary |
| | Packaging | 1. Selecting appropriate packaging materials<br>2. Avoid too heavy or too large packages. Packaging according to sterilizer manufacturer's IFU and local regulations.<br>3. Verifying the effectiveness of sterilization for new devices |
| | Loading | 1. Following guidelines and standards. Same type of material should be sterilized together.<br>2. Using absorbent tray liners to avoid wet packs<br>3. Rigid containers should be equipped with corner guards<br>4. Filling the gaps between the packs and corner protection with absorbent liner to avoid wet packs |

The $T_1$, $T_2$, $T_3$ and $T_4$ were recorded every 10 seconds throughout the sterilization process. In the formula (1), the measured sterilization temperature range was defined as the difference between the maximum recorded value and the minimum recorded value of $T_1$, $T_2$, $T_3$ and $T_4$ throughout the sterilization process. In the formula (2), the measured maintenance time was the duration from when all readings of the $T_1$, $T_2$, $T_3$ and $T_4$ first reached 134°C until the time point at which any of the four temperature sensors fell back to 134°C. In the formula (3), the difference of measured temperature uniformity was the maximum difference between the maximum and the minimum values of $T_1$, $T_2$, $T_3$ and $T_4$ recorded every 10s.

**Table 2. Other evaluation indicators of the early warning test of steam sterilization failure.**

| evaluation indicator | control group | test group | $\chi^2$ | P |
|---|---|---|---|---|
| Scheduled downtime (hours) from advance intervention | 0.00 | 3.37 | | |
| Unexpected downtime (hours) | 339.93 | 0.00 | 314.75 | <0.01 |
| Normal operation rate of the equipment | 91.34% | 99.90% | 300.92 | <0.01 |
| Overtime (hours) | 279.50 | 0.00 | | |
| The portion of overtime due to equipment failure | 37%.01 | 0.00 | | |

The overall safety margin for sterilization in this study was determined to be the minimum value obtained from the three formulas listed above. We applied the safety margin formulas across over 1000 sterilization processes, and the warning threshold was determined as the average value minus standard deviation, yielding a threshold of 72.71%. If the safe margin equaled or exceeded this threshold, an early warning would be triggered.

## Management practice

During the test, the control group processed a total of 155664 sterilization packages, with a total operation time of 3925.82 hours. Among these packages, 315 were classified as unqualified, resulting in a qualification rate of 99.78%. The control group reported 14 fault alarms from the steam sterilizer, including 3 during pre-vacuum phase, 8 during sterilization phase and 3 in other phase. Due to fault alarms, 203 packages of reusable devices needed to be reprocessed, accounting for 59.38% of the total delayed delivery of sterile items. In contrast, the test group sterilized a total of 146183 sterilization packages, with a total operation time of 3489.70 hours, achieving a qualification rate of 100%. There were no sterilization packages that required reprocessing due to alarms. The test group used 7 early warning indicators, with 3 used during the pre-vacuum phase and 4 during the sterilization phase. The main maintenance methods employed included fastening relevant pipelines and calibrating the temperature sensors. Other evaluation indicators are shown in Table 2.

## Discussion

Medical devices that enter sterile tissues, organs, cavity spaces, or come into contact with non-intact skin and mucous membranes pose a high risk of infection and therefore require sterilization [11]. Steam sterilization that uses high-temperature saturated steam is recognized as one of the most reliable, safest and cost-effective sterilization methods available [2, 3]. Inadequately disinfected medical devices increase the risk of pathogen transmission between humans and from human to environment [19]. Several studies have demonstrated a strong association between sterilization failure and the occurrence of device-related infections [20].

According to the WS310 Part 3: Surveillance *Standard for Cleaning, Disinfection and Sterilization*, continuous monitoring and recording of sterilization parameters such as temperature, pressure and time are mandatory in sterilization overshoot. Temperature fluctuations during sterilization should be controlled within a range of 3˚C, although specific requirements for the exhaust capacity in pressure and pre-vacuum phase remain unspecified [16]. While medical facilities can use the automatic control programs for steam sterilizers, most staff members lack an understanding of the parameters and principles of steam sterilization, which may lead to inadequate control of sterilization parameters and consequently increase the potential risk of failure [21]. A study evaluating the performance of steam sterilizers covering 135 medical institutions in a large city in central and western China showed that the total qualified sterilization rate was 84.07%. The qualified sterilization rate in tertiary hospitals was 93.13%, 80.26%, The

qualification rate for recorded temperature in secondary hospitals and lower-tier hospitals was 88.42%, and the qualified rate of measured temperature was only 38.95%. These findings highlight persistent issues regarding unqualified physical monitoring and significant discrepancies in sterilization effectiveness. Sterility assurance requires continuous attention to the sterilizer performance, as well as all aspects of the sterilization process [22]. Our study introduced a risk checklist to serve as early warning indicators for steam sterilization failure. We presented an expert consensus on preventing steam sterilization failure, focusing on "Staff", "Steam sterilizer" and "Medical devices". Each indicator includes feasible prevention measures derived from daily sterilization practices. For instance, under the "Steam sterilizer" category, we set three indicators: "Process monitoring", "Equipment inspection" and "Equipment maintenance". In the "Process monitoring", we added "Wet pack inspection and root cause analysis" in addition to national standards for Central Sterile Supply Department (CSSD)—Part 3: *Surveillance Standard for Cleaning, Disinfection and Sterilization* (WS 310.3–2016). For "Equipment inspection", we provided the specific checkpoints and the inspection frequencies. In the "Equipment maintenance", we provided failure prevention measures, such as necessary sensor calibration procedures under four conditions, designed to prevent steam sterilization failure induced by sterilizer failure.

Pressure, temperature and exhaust capacity are critical factors affecting the sterilization effectiveness of prevacuum steam sterilizers. Steam quality also directly influences the sterilization process [22]. The control systems of pre-vacuum steam sterilizers are only designed to uphold basic operational standards; however, if parameters exceed the preset limits, the steam sterilization process will fail [23]. In this study, we developed an early waning model, jointly designed by sterile supply experts and sterilizer engineers. This model focused on the three factors, establishing early warning indicators aimed at enabling "active" intervention in the risk of sterilization failures. The operators of sterilizers can identify potential risks by assessing different temperature and different readings of pressure sensors at various locations. Therefore, they can provide specific maintenance and adjustment at an early stage to avoid sterilization failure. This model also aims to mitigate the sterilizer shutdowns and maintenance costs caused by unqualified monitoring results reported by the control system, ensuring a consistent and reliable supply of sterile items and enhancing the overall efficiency of the CSSD.

The early warning model built in this study is a collection of management tools designed based on the genuine demands of CSSD, employing an expert consultation technique and referring to the notion of safety margin in industrial production. It utilizes warning parameters to facilitate preemptive interventions in sterilization quality. Additionally, an "intervention threshold" is tailored to the specific circumstances of each hospital, enabling local optimization of the early warning model to better accommodate the needs of various medical institutions. Safe and effective requires continuous quality improvement processes, proper staff competencies, risk assessments and proper and consistent use of all sterilization monitoring tools.

## Supporting information

**S1 Table. Physical monitoring data of sterilization process before intervention.**
(XLSX)

**S2 Table. Physical monitoring data of sterilization process after intervention.**
(XLSX)

## Author Contributions

**Conceptualization:** Xin Zhao.

**Data curation:** Xin Zhao, Ning Ma, Ran Wang, Ying Li, Tong Shen.

**Formal analysis:** Xin Zhao, Ying Li, Tong Shen.

**Methodology:** Xin Zhao.

**Project administration:** Ting Liu.

**Supervision:** Ting Liu, Tong Shen.

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
