## [Decision Letter · Decision Letter 0]

4 Oct 2024

PONE-D-24-20423Construction and practice of early warning model for failure of prevacuum pressure steam sterilization methodPLOS ONE

Dear Dr. Shen,

Thank you for submitting your manuscript to PLOS ONE. After careful consideration, we feel that it has merit but does not fully meet PLOS ONE’s publication criteria as it currently stands. Therefore, we invite you to submit a revised version of the manuscript that addresses the points raised during the review process.

We look forward to receiving your revised manuscript.

Kind regards,

Mansureh Ghavam

Academic Editor

PLOS ONE

Journal Requirements:

4. We suggest you thoroughly copyedit your manuscript for language usage, spelling, and grammar. If you do not know anyone who can help you do this, you may wish to consider employing a professional scientific editing service.  

5. Thank you for stating the following in your Competing Interests section: None. 

Reviewers' comments:

Reviewer's Responses to Questions

**Comments to the Author**

1. Is the manuscript technically sound, and do the data support the conclusions?

Reviewer #1: Yes

Reviewer #2: Partly

2. Has the statistical analysis been performed appropriately and rigorously? 

Reviewer #1: Yes

Reviewer #2: Yes

3. Have the authors made all data underlying the findings in their manuscript fully available?

Reviewer #1: Yes

Reviewer #2: No

4. Is the manuscript presented in an intelligible fashion and written in standard English?

Reviewer #1: No

Reviewer #2: Yes

5. Review Comments to the Author

Reviewer #1: The topic of the article is very relevant, the study presented is quite detailed and the techniques are compatible with the intended results, but the authors should not only review the grammar, but also some typing errors throughout the text.

In addition, there are some errors throughout the text, for example: In the Abstract the author mention that they divided the indicators into 4 categories, but they only describe 3.

On page 10, penultimate line: the indicator "very familiar" was repeated 3 times and one indicator was missing.

If the formula described is correct (Ca=(Ci+Cs)/2), then the result on the penultimate line of page 11 is not correct, despite being higher than required.

Reviewer #2: The manuscript is of paramount relevance and has high publication potential. However, there are significant adjustments that need to be made to comply with scientific standards. The main observation is about the method used and the discrepancies between the theme, the objective in the abstract, and the objective at the end of the introduction. The absence of standardized criteria and clarifications on the execution mode of each stage weakened the document. All noted observations were described in the attached document and must be followed to enhance the study's reliability, transparency, and reproducibility. Sincerely, Reviewer.

6. PLOS authors have the option to publish the peer review history of their article (what does this mean?). If published, this will include your full peer review and any attached files.

Reviewer #1: No

Reviewer #2: **Yes: **Rebecca S. C. Santos

---

## [Author Response · Author response to Decision Letter 0]

25 Nov 2024

Dear editor,

Thank you for correcting the article "Construction of management tools for early warning of prevacuum steam sterilization failure". We made targeted changes and enhancements to the article after carefully reviewing expert revision opinions, and we have noted the precise changes in the article for your evaluation. Now, the revision ideas and issues raised by the review specialists are described one by one.

Reviewer 1: 

Comment 1: The topic of the article is very relevant, the study presented is quite detailed and the techniques are compatible with the intended results, but the authors should not only review the grammar, but also some typing errors throughout the text.

Response 1: Thank you for you comment, we had sent the manuscript to a professional paper polishing agencies, to check grammatical errors and optimize the quality of language expression

Comment 2:In addition, there are some errors throughout the text, for example: In the Abstract the author mention that they divided the indicators into 4 categories, but they only describe 3.

Response 2: Thank you for your comment, we had revised this mistake and checked the consistency of the preceding text.

Comment 3: On page 10, penultimate line: the indicator "very familiar" was repeated 3 times and one indicator was missing.

Response 3: Thank you for your comment, we had revised this mistake.

Comment 4: If the formula described is correct (Ca=(Ci+Cs)/2), then the result on the penultimate line of page 11 is not correct, despite being higher than required.

Response 4: Thank you for your comment, since the questionnaire for expert had been revised in this study, we confounded the results before and after adjustment. We had revised this mistake, and all the statistical results of the questionnaire were uniformly taken as three decimal places.

Reviewer 2: 

Comment 1: I suggest replacing “Construction and practice [...]” with “Construction and clinical validation [...]”. The purpose of the substitution is to clarify the study method employed.

Response 1: We had revised the paper title.

Comment 2: The objective is incomplete; it is evident that a mixed-methods study was employed, apparently a methodological study followed by a content validation and clinical validation. Clarify this. The employed method also lacks further explanations, even if they seem obvious. What is the study design? How many and what were the phases of development? Based on what methodological framework? What do you consider “specialized consultation”? The results indicate six alert indicators in four categories; this is incomplete, write them out in full, both the indicators and the categories.

Response 2:We have revised the Objectives and methods section of the abstract to specify the objectives and methods of the study. The main objectives of the study, including the management tools for the expected construction and the controlled trials, are introduced in the objectives, and the phases of the study and the content planned to be completed in each phase are introduced in the Methods of the abstract. The presentation of indicators and categories was also corrected

Comment 3: Insert two more descriptors with greater potential for generalization on the subject.

Response 3: We have added three keywords, namely Management tool; Safety margin; Expert consultation

Comment 4: Insert the state of the art on the subject, the justification, and the relevance of the study. The inserted objective deviates from what is mentioned in the summary and needs standardization, even if it seems obvious to you.

Response 4: We systematically adjusted the background section, adding new content and references around the recent progress of the research topic and the need for research,

Comment 5: The study design does not seem clear within the various types of studies that exist and needs to be revised. The methodological reference model used is not mentioned, and it is necessary to follow one for scientific standardization and methodological rigor. The number of phases is implied; make it evident. The recruitment process for specialists and their function is not clear. A data search for identifying the indicators was mentioned; however, the process mentioned earlier was the interview with specialists. Which is the correct method? The terms “control group” and “experimental group” are commonly used for experimental studies, and are not suitable for this study, evaluate the possibility of replacing it with a similar one.

Response 5:In this study, expert consultation method was used as the main method to reach a multidisciplinary consensus on the failure risk checklist and the calculation model of early sterilization failure risk. The failure risk checklist was selected and included inspection items unanimously approved by experts based on the results of previous studies. The calculation model of early sterilization failure risk was based on the margin of safety theory in industrial manufacturing, and a consensus was reached on the calculation formula for early warning of sterilization failure using the temperature and pressure sensor readings of pressure sterilizers. In addition, we changed the names of "control group" and "experimental group" to "test group" and "conventional group" during the method validation process.

Comment 6: Nine indicators appeared, differing from what is stated in the abstract; minor adjustment needed.

Response 6: Thank you for pointing out the error in this article, we have corrected it

Response 7: Create a pamphlet regarding the conclusions of your study.

Response 7: We systematically adjusted the conclusions and added summary statements

---

## [Editor Report · Decision Letter 1]

11 Dec 2024

Construction of management tools for early warning of prevacuum steam sterilization failure

PONE-D-24-20423R1

Dear Dr. Shen,

We’re pleased to inform you that your manuscript has been judged scientifically suitable for publication and will be formally accepted for publication once it meets all outstanding technical requirements.

Kind regards,

Mansureh Ghavam

Academic Editor

PLOS ONE
---

## [Editor Report · Acceptance letter]

14 Dec 2024

PONE-D-24-20423R1 

PLOS ONE

Dear Dr. Shen, 

I'm pleased to inform you that your manuscript has been deemed suitable for publication in PLOS ONE. Congratulations! Your manuscript is now being handed over to our production team.

Kind regards, 

on behalf of

Dr. Mansureh Ghavam 

Academic Editor

PLOS ONE